# Response of Castor Seedling Roots to Combined Pollution of Cd and Zn in Soils

Feifei Wang [1],[†] , Linlin Yang [1],[†], Yanping Zhao [1], Zhenzhen Zhao [1], Kokyo Oh [2],* and Chiquan He [1],*

[1]    School of Environmental and Chemical Engineering, Shanghai University, Shanghai 200444, China
[2]    Center for Environmental Science in Saitama, 914 Kamitanadare, Kisai, Saitama 347-0115, Japan
*    Correspondence: o.kokyo@pref.saitama.lg.jp (K.O.); cqhe@shu.edu.cn (C.H.)
†    These authors contributed equally to this work.

**Abstract:** Castors are used to remediate heavy-metal-polluted soils due to their good ability to accumulate heavy metals. However, only limited studies addressed the interaction between heavy metals and castor seedling roots. In this study, the physiological response of castor seedling roots to Cd and Zn stress, and the change in trace elements in rhizosphere and non-rhizosphere soils were investigated. The results showed that, with an increase in Cd concentration, the accumulation of Zn in roots decreased by 20%, indicating a competition between Cd and Zn accumulation. The increase in Cd content enlarged the difference in nutrient contents at different depths: the amounts of P, Fe, and Mn were more in rhizosphere soils than in non-rhizosphere soils, while the amount of K showed an opposite trend. The addition of Cd and Zn stimulated root growth, but root activity was reduced. The addition of Cd and Zn affected the root cell morphology, including increases in the root cortex thickness and the root xylem area. The contents of the enzymes SOD, POD, and MDA increased with the addition of Cd and Zn, while the activity of CAT first increased and then decreased. There was no significant change in the soluble protein content. The decrease in IAA oxidase content, from 40.1% to 72.7%, was conducive to plant growth. To sum up, high contents of Cd and Zn in soils not only affect the root morphology and increase the gap in the contents of K, P, Fe, and Mn between rhizosphere and non-rhizosphere soils but also change the SOD, POD, MDA, and IAA contents in the root, so as to reduce the amount of root damage caused by the external environment.

**Keywords:** accumulation; tolerance; heavy metal; castor seedling roots; physiological response





## 1. Introduction

The heavy metal contamination of soils is a serious environmental problem due to the toxicities of these metals to both humans and living organisms. Excessive cadmium (Cd) and zinc (Zn) induce serious risks to human health through contamination of the food chain and of drinking water [1]. Therefore, more and more researchers have paid attention to heavy metal pollution due to this widespread concern. The Ministry of Land and Resources Report issued a national soil contamination survey and reported that 16% of the Chinese soil samples surveyed, consisting of 6.3 million $km^2$, were contaminated [2]. A previous study investigating 5000 soil samples from 1781 farmlands reported that the average concentration of Zn in the farmland soils of China is 4.7 times higher than the secondary environmental quality standard for soils [3]. Phytoremediation is accepted by researchers as a green, environmentally friendly, and economical method of remediation [4]. The energy crop castor, *Ricinus communis*, is considered a potential extractive plant because of its strong vitality, its large biomass, its high extraction of heavy metals, and its prevention of heavy metals from entering the food chain [5–7]. Castor is an important oil seed crop worldwide, and no vegetative part of castor is consumed by human beings or cattle. Therefore, it can be safely used for soil phytoremediation [8]. However, to the best of our knowledge, there are very few reports addressing an enhancement in the phytoremediation potential of castor

bean plants [9]. Cd and Zn are both divalent metal elements, and high concentrations cause negative effects on both plant and human health [10]. Its adverse effects on plants involve destroying the cell structure [11], affecting antioxidant enzyme systems [12], and inhibiting photosynthesis and plant growth.

Plant roots can support the aboveground parts. It plays an important role in material storage, in the absorption of nutrients such as water and inorganic salts, and in the synthesis of substances such as amino acids and hormones. The aboveground part and the underground part of the plant are interdependent for material exchange, energy transfer, and information exchange within the plant. Generally speaking, the more developed the root system, the better the aboveground part can grow. In the phytoremediation of heavy-metal-contaminated soils, many studies have shown that the heavy metals accumulated are mainly concentrated in the roots [13–15]. Heavy metals are adsorbed on the surface of the root system and then enter the root through the root cortical tissues, along with water; nutrient salts; and their transporters, involving the iron-regulated transporter-like protein (ZIP) family, zinc-regulated transporters, and the heavy metal ATPase transporters (HMA) [16,17]. The heavy metals enter the plant; are loaded into the xylem; and are then transported to the ground part, driven by transpiration and root pressure. In heavy-metal-contaminated soils, plant roots come into direct contact with heavy metal pollutants and are damaged to varying degrees, such as plasma membrane damage and reactive oxygen species damage [18,19]. One of our previous studies found that the sequences for Zn and Cd accumulation in castor seedling tissues were root > leaf > stem and root > stem > leaf, respectively, indicating that castor seedling roots play important roles in accumulating Zn and Cd [20]. Therefore, the root system is an important part in heavy metal enrichment and reduction in heavy-metal stress. However, there are very few studies addressing the interaction between heavy metals and castor seedling roots. It is necessary to study the effect of heavy metals on roots of castor seedlings.

The physiological response of plants to Cd and Zn is important for explaining the stress mechanism. In plants, one of the important reasons why heavy metals are toxic is the indirect induction of reactive oxygen species (ROS), including superoxide ($O_2^-$) and hydrogen peroxide ($H_2O_2$) [15]. Active oxygen in plants is produced by the combined action of photosynthesis in chloroplasts, the aerobic respiration in mitochondria, and the photorespiration in peroxisomes [21]. Under the stress of heavy metals, plants produce excessive ROS, which react with lipids and proteins, resulting in cell-membrane damage [22]. To reduce the occurrence of oxidative damage, the organism produces antioxidant enzymes, including superoxide dismutase (SOD), guaiacol peroxidase (POD), catalase [23], and ascorbate peroxidase (APX) [24]. Superoxide radicals are converted into $H_2O_2$ and oxygen ($O_2$) by SOD, and then, $H_2O_2$ is reduced to $H_2O$ and $O_2$ by CAT, POD, and APX in the cytoplasm and in other cells [25]. These enzymes work together with other enzymes to maintain and regulate the level of active oxygen in the plant at a relatively low level for normal physiological activities. Therefore, the defense system produced by antioxidant enzymes is an important mechanism for plants to resist heavy-metal stress.

Considering the aforementioned information, it is proposed that direct contact of castor seedling roots with heavy metals may affect early root cell morphology development and that the roots of castor seedlings may produce antioxidant enzymes to resist heavy-metal stress to ensure normal growth. Therefore, the purpose of this study was to investigate the effects of castor seedling roots on the accumulation of Cd, Zn, and other trace elements, and the morphological and physiological responses of castor seedling roots to combined Cd and Zn pollution.

## 2. Materials and Methods

### 2.1. Chemicals and Preparation

Castor seeds used in the pot experiment were purchased from the online seed store (Yucaoyangshengtang, hefei, China), and the experimental soils were taken from a non-polluted farmland at the campus of Shanghai University (Shanghai, China). The collected

soil was air-dried naturally and passed through a 10 mesh sieve (0.25 mm) before use. Approximately 5 g of soil was collected to determine the background values of the soil, including a pH of 8.03, a TN of 1.69, a TP of 0.93, and a TK of 19.2. Additionally, after soil aging, the concentrations of Zn and Cd in the soils were measured, shown in Table 1.

**Table 1.** The contents of Zn and Cd experimental soil.

| Physical or Chemical Parameters | Soil Background | Amount of Heavy Metals Added (mg/kg of Soil) | After Adding Heavy Metals |
|---|---|---|---|
| Cd | 0.18 mg/kg | 25<br>5<br>0.5<br>0 | 23.54 mg/kg<br>5.4 mg/kg<br>0.46 mg/kg<br>0.2 mg/kg |
| Zn | 73.25 Mg/kg | 380<br>0 | 448.1 mg/kg<br>93 mg/kg |

### 2.2. Plant Culture and Experimental Design

The pot experiment was carried out for almost 3 months including heavy metal aging for 2 months and castor growth for 19 days (the longer the aging time, the more stable the form of heavy metals) [26]. The aged soils were divided into 15 flowerpots to achieve 1.5 kg soils per pot. In order to simulate the actual pollution concentration of Zn in the soils, the concentration after the addition of $ZnCl_2$ into the 12 pots was set as 500 mg Zn/kg soils [27,28]. The other 3 pots with only soils and not any extra metal addition was the control group, used as a reference. Afterwards, in the 12 pots with 500 mg of Zn/kg of soil, $CdCl_2$ was added to reach different concentrations—0, 0.5, 5, and 25 mg of Cd/kg of soil to cover the whole range from below the secondary environment quality standard for agricultural land (0.6 mg/kg) to heavy pollution. Therefore, the five treatment groups were set: control, Cd0Zn380, Cd0.5Zn380, Cd5Zn380, and Cd25Zn380. The contaminated soils in the 15 pots were aged indoors for two months. The castor seeds were soaked in 3% $H_2O_2$ for 30 min, and then rinsed with tap water and distilled water in turn. The washed castor seeds were then dried for later sowing. The castor seeds were planted on 1 September. All of the pots filled with soils with and without the heavy metals were placed near the laboratory windows for normal natural light exposure during the experiment. The harvest occurred on 20 September. The harvested seedlings were cut into roots, stems, and leaves, which were stored in a refrigerator at −80 °C for subsequent analysis.

### 2.3. Determination of Soil Properties

The contents of Cd, Zn, and some nutrient elements (K, P, Fe, Cu, and Mn) in non-rhizosphere soils (0 cm) and rhizosphere soils (5 cm) were tested after natural air drying. After 0.1 g of the soils was digested with $HNO_3$ + $HClO_4$ + HF and clarified, the concentrations of Zn and some nutrient elements (K, P, Fe, Cu, and Mn) were determined by inductively coupled plasma (ICP, Leeman, WI, USA), and the content of Cd was determined by a graphite furnace atomic absorption spectrophotometer (AAS, Jena, LA, USA).

### 2.4. Root Index Determination

On harvest day, the seedlings were carefully collected without damaging the castor body. Immediately, the seedling roots were rinsed using tap water and distilled water, in order. Afterwards, the root length, amount, and weight were measured.

The root vigor determines the alpha-naphthylamine oxidation. The plant root system has an oxidation effect on α-amine due to the breathing effect of plants. Peroxidase plays an important role in α-amine oxidation. The stronger the vitality of the enzyme, the stronger the oxidation of α-amine and the higher the plant root system vitality. Root dynamics are closely related to plant nutrient absorption, metabolism, and plant growth. The stronger the root dynamics, the better plant growth is. A total of 2 g of a fresh seedling root was

placed in an Erlenmeyer flask filled with 50 mL of a mixed solution of alpha naphthylamine with a concentration of 50 μg/mL and phosphate pH buffer solution (pH = 7.0). The root immersed in the solution was shaken gently and left to stand for 10 min. Afterwards, 2 mL of the solution was collected to analyze the concentration of alpha naphthylamine. Subsequently, the triangular bottle was plugged into a 25 °C incubator and the content of n-naphthylamine in the solution was measured every 2 h. In addition, the group without root treatment was set as the control.

### 2.5. Cd and Zn Content in Roots

The seedlings were rinsed with tap water and distilled water in turn before being divided into roots, stems, and leaves. They were dried in an oven at 75 °C and then ground into a fine powder; then, 0.1 g of the powder was taken and digested using $HNO_3/HClO_4$ (5:1, *v/v*). After the above pre-treatment, the amount of Zn was determined by inductively coupled plasma atomic emission spectrometry (ICP-OES, Leeman, WI, USA) and the amount of Cd was determined by graphite furnace atomic absorption spectrophotometer (AAS, Jena, LA, USA).

### 2.6. Root Cross-Sectional Morphology

To observe the effect of Cd on root morphology, castor seedlings were pulled out carefully without damaging the root structure and then a small piece of the cross section with a thickness of 3 mm was cut from 3 mm below the top of the taproot for subsequent detection by microscopy. The root tissues harvested from the three treatment groups (control, Cd0Zn380, and Cd25Zn380) were fixed with a formaldehyde-acetic acid-ethanol fixative (FAA) fixative for one day, and then, the root tissues were evacuated. The fixed root tissues were dehydrated with gradient ethanol and embedded in paraffin. The paraffin-embedded samples were cut into 10 μm using a rotary microtome, and the cut samples were stained using toluidine blue. The structural characteristics of the stained sample were observed with a light microscope (ZEISS Axio Scope.A1, Oberkochen, Germany) at a magnification of 25×.

### 2.7. Antioxidant Enzymes and Physiological Indicators

A crude enzyme extract was prepared according to the method used: 0.5 g of the fresh root samples were washed and placed in a precooled mortar, and 50 mmol/L of the precooled phosphoric acid buffer (pH = 7.8) was ground in an ice bath to form a homogenate. The supernatant was then centrifuged at $12,000 \times g$ at 4 °C for 20 min, and the supernatant as the enzyme solution was collected for subsequent enzyme measurement [29].

For the SOD measurement, 30 μL of the enzyme solution and 3 mL of the reaction solution containing 0.05 M phosphate-buffered solution (PBS), 14.5 mM methionine, 30 μM disodium ethylenediamine tetracylate, 60 μM riboflavin, and 2.25 mM azolantethazole were mixed and incubated in a light incubator at 4000 lux for 20 min. Afterwards, the absorbance was determined at 560 nm.

For the POD measurement, 3 mL of the reaction solution containing 1.886 mL of 0.2 M PBS, 0.459 mL of 2-methoxyphenol, and 0.765 mL of 30% $H_2O_2$ was mixed with 30 μL of the enzyme solution. The absorbance of the reaction solution was determined at 470 nm.

For the CAT measurement, 3 mL of the reaction solution containing 0.15 M PBS and 30% $H_2O_2$ was mixed with 0.1 mL of the enzyme solution. After 2 mins of the reaction, the absorbance was determined at 240 nm. The soluble protein content was determined by Coomassie blue staining. The absorbance was determined at 595 nm after 2 min of a reaction of 20 μL of the enzyme solution and the reaction mixture (0.05m PBS, pH = 7.8, Coomassie bright blue).

Malondialdehyde (MDA) was detected using the modified thiobarbituric acid method [30]. First, fresh plant root samples (2 g) and 2 mL of the 10% trichloroacetic acid solution were ground in an ice bath until homogenization; next, it was centrifuged at $12,000 \times g$ for 10 min; then, 1.5 mL of the supernatant and the 0.67% thiobarbital acid

solution each were boiled together in a boiling water bath for 30 min; and finally, following cool down and centrifugation, the supernatant's absorbance was measured at three wavelengths: 450, 532, and 600 nm.

For the IAA oxidase measurement, 0.5 g of the fresh seedlings collected from the rhizosphere were added into 5 mL of the pre-cooled phosphate buffer, then ground until homogenization under an ice bath, and centrifuged at $4000 \times g$ for 20 min. The supernatant was the crude enzyme solution. Afterwards, 1 mL of the 1 mM manganese chloride solution, 1 mL of 1 mM dichlorophenol, 2 mL of 1 mM IAA, and 5 mL of the phosphate buffer were added to 1 mL of the enzyme solution one by one. In addition, the phosphate buffer was used instead of the enzyme solution in the control group, and the other reagents added did not change. The test tube was placed in a water bath with a constant temperature of 30 °C for 30 min. Then, 2 mL of the above reaction mixture was absorbed. Subsequently, 4 mL of indoleacetic acid reagent B (0.5 M $FeCl_3$, 350 g/L perchloric acid) was added into the 2 mL mixture solutions above. Then, it was placed in the dark at 30 °C for 30 min. After color development, the wavelength of the reaction liquid was determined at 530 nm.

### 2.8. Statistical Analysis

The data were analyzed by two-way ANOVA with Duncan's test ($p < 0.05$) using SPSS 19 for Windows (SPSS Inc., IBM Corporation, Armonk, NY, USA). All of the data are expressed as the mean $\pm$ SD of three replicates. The differences between different treatments were calculated using the least significant different test. All of the figures were produced using Origin Pro 9, OriginLab, Northampton, MA, USA.

## 3. Results

### 3.1. Cd and Zn Accumulation in Castor Seedling Roots

The cumulative concentrations of Cd and Zn in the roots are shown in Figure 1. Compared with the control group, the maximum accumulation of Zn (386.8 mg/kg) was found in the Cd0Zn380 treatment group (Figure 1A), and the addition of Zn inhibited the roots from absorbing Cd (Figure 1B). However, in the Cd and Zn addition groups (Cd0.5Zn380, Cd5Zn380, and Cd25Zn380), the cumulative concentration of Zn in the roots did not change evidently with the increase in Cd content from 0.5 to 25 mg/kg. The accumulation of Cd in the roots increased nonlinearly with the increase in Cd content. In the group of Cd25Zn380, the concentration of Cd in the roots reached as high as 175.3 mg/kg. Overall, the results imply that there was competition between the accumulation of Cd and Zn in castor seedling roots.

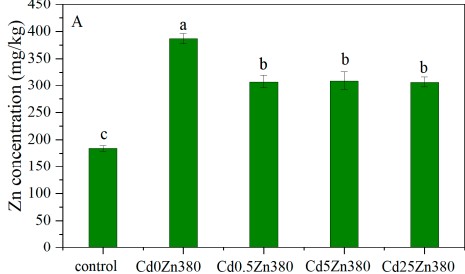 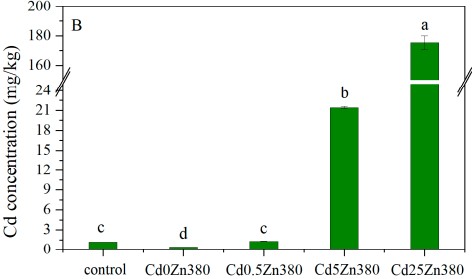

**Figure 1.** The accumulation of Cd (**A**) and Zn (**B**) in castor seedling roots under different heavy-metal pollution concentrations. Different letters, a–d, indicate significant differences ($p < 0.05$).

### 3.2. Effects of Cd and Zn on Soil Nutrient Elements

Figure 2 presents the concentrations of K, P, Fe, and Mn in non-rhizosphere and rhizosphere soils in the groups with different heavy-metal contents. It can be observed that all P, Fe, and Mn concentrations in the rhizosphere soils did not show evident differences among different treatment groups while all decreased with the increase in Cd concentration in non-rhizosphere soils. K concentration in non-rhizosphere soils changed a little in all treatment groups, while K concentration in rhizosphere soils significantly decreased in the

Cd25Zn380 group. P, Fe, and Mn all showed higher concentrations in rhizosphere soils than in non-rhizosphere soils. It is noteworthy that, in the Cd25Zn380 treatment group, the concentrations of P, Fe, Cu, and Mn in the non-rhizosphere soils were significantly lower than those in the rhizosphere soils, which might be explained by the mass flow response of plants to regulate nutrient migration in order to maintain normal growth [31].

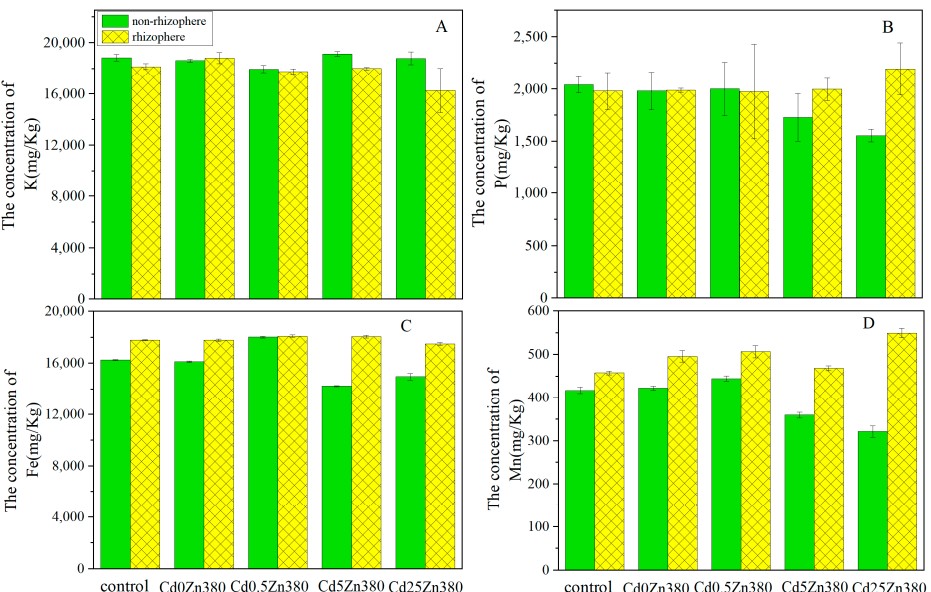

**Figure 2.** The concentrations of K (**A**), P (**B**), Fe (**C**), and Mn (**D**) in non-rhizosphere and rhizosphere soils in different treatment groups.

The differences in nutrient element contents between the rhizosphere and non-rhizosphere soils were also observed in previous studies. Sato et al. [32] reported that the use of a 0–3 cm thick revegetation netting can regulate the migration of cesium in deeper soils to surface soils to remove pollutants in soils [32]. Rhizosphere soil has more active microbial activities, which affect enzyme activities and root respiration in the rhizosphere soils, which can then regulate the distribution of nutrients in the soils [33,34].

*3.3. Morphological Response of Castor Seedling Roots to Cd and Zn*

In order to evaluate the effects of Cd and Zn on castor seedling root growth, the castor seeding morphology and enzyme activity were analyzed. In Figure 3A,B, the length and number of roots first increased and then decreased with the increase in Cd content. In the Cd0.5Zn380 treatment group, the root length and number showed the maximum values, which were 11 cm and 23, respectively. Figure 3C shows that the fresh root weight of castor seedlings also first increased and then decreased with the increase in Cd content. The fresh root weight in all treatment groups were higher than the control group, and the largest biomass was 0.54 g in the Cd5Zn380 group. These results suggest that low concentrations of Cd and Zn could promote root growth in the castor seedlings, while high concentrations would inhibit root growth.

Figure 3D showed the change in α-naphthylamine content in the roots of different treatment groups. The α-naphthylamine content reflects the change in root vigor. Compared with the control group, the content of α-naphthylamine in all treatment groups decreased significantly, indicating that the root vigor of treatment group was significantly lower than that of the control group. The addition of Zn alone significantly decreased the α-naphthylamine content, and the simultaneous addition of Zn and Cd further decreased the α-naphthylamine content, implying that both Zn and Cd are able to lower the root vigor of castor seedlings.

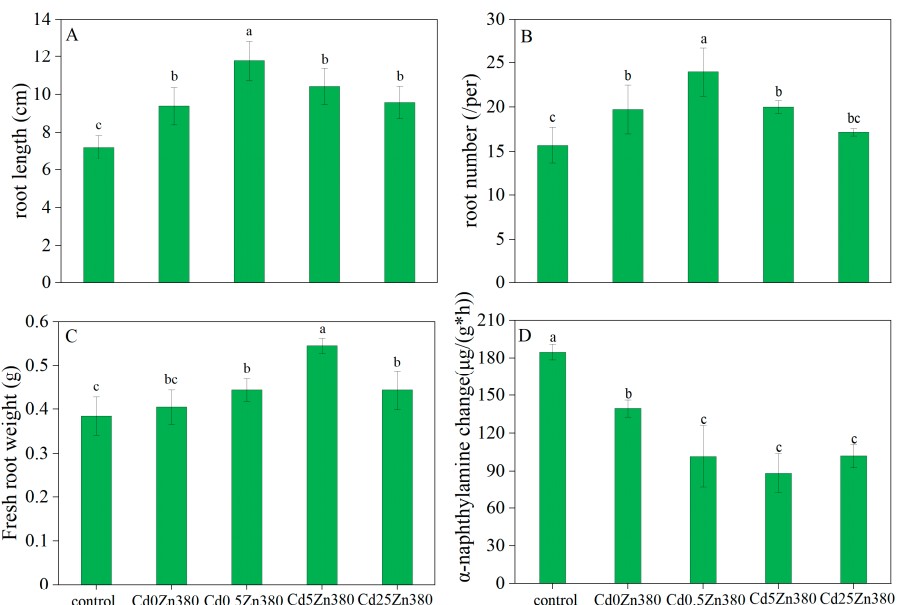

**Figure 3.** Effects of Cd and Zn on root morphology including root length (**A**), root number (**B**), fresh root weight (**C**), and root activity (**D**). Different letters, a–c, indicate significant differences ($p < 0.05$).

### 3.4. Micromorphology of Castor Seedling Roots to Cd and Zn

Figure 4 shows the cross sections of castor seedling roots at 25 and 100 times magnification in three treatment groups (control, Cd0Zn380, and Cd25Zn380). The cross sections of castor seedling roots in the control group showed an oval shape, which is normal, while Cd0Zn380 and Cd25Zn380 showed irregular shapes, which are more disorderly. Compared with the control, the lateral roots can be clearly observed in the Cd0Zn380 and Cd25Zn380 treatment groups, which is consistent with the abovementioned results. The cross sections of the castor seedling roots are mainly composed of cortex, cambium, and xylem. The cortex of the roots was thicker in the Cd0Zn380 and Cd25Zn380 groups than in the control, indicating that the cortex was stimulated under Cd and Zn stress. It is known that the cortex is the main accumulation site for heavy metals in the root [35]. Therefore, a thicker cortex might be an adaption for the accumulation of Zn and Cd in seedling roots. The xylem is the transport organ in vascular plants and is responsible for transporting water and the ions dissolved in the water. The change in xylem may be due to the response to Cd and Zn stress in castor seedlings. Moreover, by comparing the Cd0Zn380 and Cd25Zn380 treatment groups, the xylem proportion of the latter decreased, which might be due to Cd stress. These results suggest that, after adding Cd and Zn, the castor seedling roots responded to Cd and Zn stress mainly by changing the cellular structure in the roots.

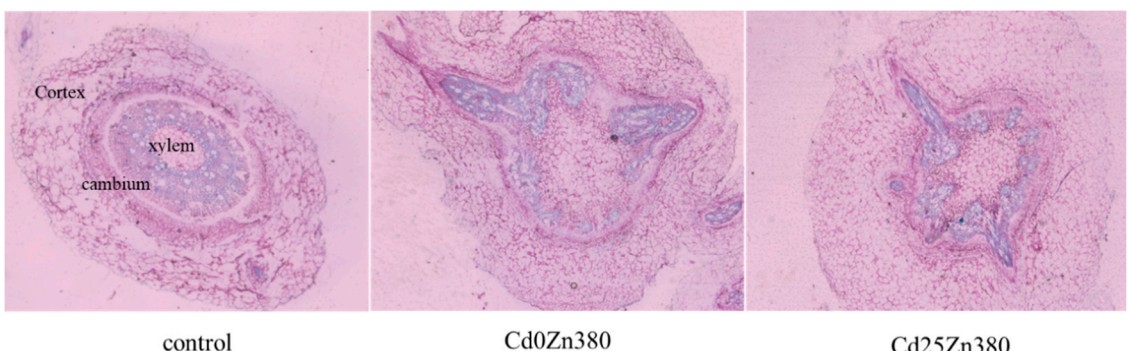

**Figure 4.** Cross-sectional diagrams of castor seedling roots in three treatment groups: control, Cd0Zn380, and Cd25Zn380.

### 3.5. Physiological Response of Castor Seedling Roots to Cd and Zn

Figure 5 shows the physiological changes in the castor seedling roots in response to Cd and Zn stress. The SOD enzyme activity of the treatment groups was significantly higher than that of the control group (Figure 5A). With the increase in Cd concentration, SOD enzyme activity first increased then decreased. The CAT content in the Cd0Zn380 treatment group was significantly higher than that in the control, and the CAT enzyme activity decreased significantly with the increase in Cd content. The changes in SOD, POD, and CAT suggest that, under Cd and Zn stress, castor seedling roots could resist the stress by regulating the activity of antioxidant enzymes.

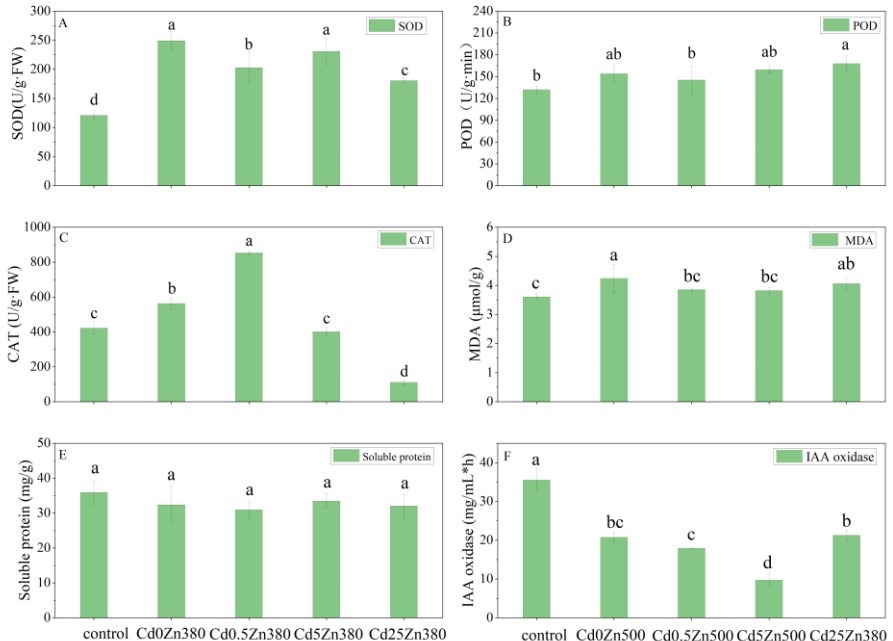

**Figure 5.** Physiological response of castor seedling roots to Cd and Zn. SOD (**A**), POD (**B**), CAT (**C**), MDA (**D**), problem protein (**E**) and IAA oxidase (**F**). Different letters indicate significant differences ($p < 0.05$).

The content of MDA in the control group was lower than that in the treatment groups. As the Cd content increased, the MDA content also increased. The soluble protein content of the roots did not change significantly. The activity of IAA oxidase in seedlings in the treatment group was lower than that in control group. With the increase in Cd concentration in soils, IAA oxidase in the roots showed a trend of first decreasing and then increasing. This suggests that high levels of Cd and Zn pose a threat to the physiological activity of castor seedling roots. Castor can control and reduce the content of IAA oxidase to ensure the normal growth of plants.

## 4. Discussion

In many studies within the literature, it has been reported that the root growth of plant seedlings is inhibited to varying degrees under the stress of heavy metals [17,36,37]. It was found in this study that, under Cd and Zn stress, the root length (Figure 3B), root number (Figure 3C), and fresh root weight (Figure 3D) of castor seedlings increased compared with those in the control group. This is contrary to a previous study reporting that the root growth of plant seedlings was inhibited to different degrees under the stress of heavy metals. Therefore, castor seedlings have good tolerance to Cd and Zn stress, which proves that castor is a promising repair plant.

Root vitality is a comprehensive reflection of the number of living cells in the root and its metabolic intensity [38]. Liu et al. [39] reported that the root vigor of *Salix variegate* decreased with an increase in Cd content [39]. In Figure 3D, the root vigor of castor

seedlings was significantly reduced when subjected to Cd and Zn stress, which is consistent with the results reported in the literature. Therefore, the decrease in root vigor proves that the roots of castor seedlings are negatively affected.

An interesting phenomenon was found in this study, as shown in Figure S1 (Supplementary Materials). Castor seedling roots grew vertically downward in the control group, and they tended to spread horizontally in the treatment groups. The taproots were obvious, and the lateral roots were fewer in the control group. In treatments, the lateral roots were more developed and the growth of the taproots was inhibited. In the high-concentration group, Cd25Zn380, the taproot even disappeared.

In summary, Cd and Zn pollution enlarged the differences in the concentrations of nutrients between rhizosphere and non-rhizosphere soils. This difference could be explained by the changes in the plant root growth trends and more lateral root derivation. Additionally, the derivation of more lateral roots was to extract more nutrients. This also indicates that castor seedlings adapt to heavy-metal-pollution stress by changing root growth.

There are two processes regarding the transfer of heavy metals: longitudinal transfer from underground parts to aboveground parts, and transverse transfer from the xylem to the pith and cortex [40]. It has been reported that heavy metals accumulate in root tissues, mainly in the cortical cell wall, but rarely in the xylem cell wall [41]. One of our previous studies found that the sequences of Zn and Cd concentrations in castor seedling tissues were root > leaf > stem and root > stem > leaf, respectively, indicating that castor seedling roots play important roles in accumulating Zn and Cd [20]. In this study, Cd and Zn were mainly accumulated in the roots; compared with the control, the treatment group had some changes in the cross-cutting cell morphology of their castor seedling roots, such as the epidermis becoming thicker and the xylem becoming larger. The thickening of the phloem layer provides more binding sites for the accumulation of Cd and Zn. Studies showed that Cd accumulates mainly in the epidermis of the Sedum alfredii at the seedling stage [42]. In this study, the thickening of the epidermis may be a morphological regulation in response to stress. The xylem is mainly responsible for the plant to transport nutrients and water. An enlarged xylem can deliver more nutrients and moisture, which may be a response of the castor for maintaining normal growth and for relieving the stress from heavy metals.

Plants can use antioxidant enzymes to mitigate the physiological damage caused by heavy metal toxicity [43]. The stimulation of heavy metals can promote the production of ROSs, which leads to an increase in the content of antioxidant enzymes in plants [44,45]. The SOD enzyme activity in the treatment groups was significantly higher than that in the control group, suggesting that the addition of Cd and Zn stimulates the antioxidant enzyme system of castor seedlings to cope with the root damage caused by the external environment. The accumulation of MDA causes serious damage to cell membrane and cells. The content of MDA increased with the increase in heavy metal concentration, indicating that membrane lipid peroxidation caused by the stress of heavy metals was significant, which led to damage to the cell membrane and an imbalance in the intracellular osmotic pressure [46]. In addition, MDA can produce toxic effects on chloroplasts, mitochondria, and other organelles and can inhibit the growth of plants [47]. In this study, with an increase in the Cd content, the MDA content also increased, suggesting that the addition of Cd and Zn may damage the cells in the roots of castor seedlings and that the morphological changes in the cells of the roots may be related to this. The content of soluble protein is one of the important indexes used to measure plant stress caused by heavy metals that combine with other compounds to form metal complexes or chelates, which can inhibit various protein synthesis. In this study, there was no significant change in the soluble protein content, which was similar to the results in the literature [48]. This indicates that the presence of Cd and Zn at this concentration (Zn, 380 mg/kg; Cd, 0–25 mg/kg) had no effect on the synthesis of soluble proteins. The activity of IAA oxidase is related to stable roots and regulates their growth [49]. It has been reported that Al treatment reduced the

activity of IAA oxidase in plants, which is conducive to the growth of plants [50]. The results in this study were similar to those in the literature.

## 5. Conclusions

Our results showed that, with an increase in Cd pollution concentration, the accumulation of Zn in roots decreased, indicating a competition between Cd and Zn accumulation. The increase in Cd content enlarged the difference in nutrient contents at different depths: the amounts of P, Fe, and Mn were more in rhizosphere soils than in non-rhizosphere soils, while the amount of K showed an opposite trend. Compared with the control, the root length, number, and fresh weight showed that the addition of Cd and Zn was beneficial to the growth of castor seedling roots, but the activity of the castor seedling root was reduced. The addition of Cd and Zn affected the cell morphology of the root system of castor bean seedlings, including a thickening of the cortex and an increase in the area of the xylem in the root. The contents of SOD, POD, and MDA increased with the addition of Cd and Zn, while the activity of CAT first increased and then decreased. There was no significant change in the soluble protein content. An decrease in IAA oxidase content, that is, an increase in IAA level, is conducive to plant growth. It is suggested that high levels of Cd and Zn pose threats to the roots of castor seedlings. Castor seedlings can control and reduce the content of IAA oxidase to ensure the normal growth of plants.

**Supplementary Materials:** The following are available online at https://www.mdpi.com/article/10.3390/su141710702/s1, Figure S1. Root morphology of castor seedling roots in different treatment groups.

**Author Contributions:** Conceptualization, C.H.; investigation, Y.Z.; resources, K.O. and C.H.; data curation, Z.Z.; writing—original draft preparation, L.Y.; writing—review and editing, F.W.; supervision, C.H.; project administration, C.H. and F.W.; funding acquisition, C.H. and K.O. All authors have read and agreed to the published version of the manuscript.

**Funding:** This study was financially supported by the Shanghai Science and Technology Project (No. 19DZ1205202), Natural Science Foundation of China (No. 41971055), JSPS KAKENHI (No.16H05633) and State Key Laboratory of Pollution Control and Resource Reuse Foundation (No. PCRRF19003).

**Institutional Review Board Statement:** Not applicable.

**Informed Consent Statement:** Not applicable.

**Data Availability Statement:** Not applicable.

**Conflicts of Interest:** The authors declare no conflict of interest.

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
