# Peer review of "Response of Castor Seedling Roots to Combined Pollution of Cd and Zn in Soils"

_sustainability, doi:10.3390/su141710702_

Round 1

Reviewer 1 Report

As general comment the work is well written and designed with relevant results.

In general terms the topic of the article is interesting, the methodology is explicitly presented and the results reported are interesting.

The structure of the paper is correct.

In my opinion, the abstract is too general, please reframe.

The introduction chapter should end with a paragraph indicating the purposefulness of the conducted research. Authors should clearly define the purpose of the work and formulate research hypotheses.

Materials and method section is well described and correspond to the aim set out in the manuscript. The figures clearly presenting the obtained results with their appropriate interpretation.

How was the statistical calculations done?

The references are sufficient and necessary.

The paper needs some editorial corrections.

I recommend the publication of this manuscript in the Sustainability journal after minor revisions.

Author Response

As general comment the work is well written and designed with relevant results.

In general terms the topic of the article is interesting, the methodology is explicitly presented and the results reported are interesting. The structure of the paper is correct.

Response: The comments from Reviewer #1 are appreciated. Please note: line numbers in the Reviewer’s comments refer to the last version of the manuscript, while the line numbers in the Authors’ answers refer to the current revised manuscript.

Comment 1:

In my opinion, the abstract is too general, please reframe

Response: Thanks for this comment. The abstract has been modified in lines 18 and 26-27, respectively.

“Results showed that with the increase of Cd concentration, the accumulation of Zn in roots de-creased by 20%”

“The decrease of IAA oxidase content, from 40.1% to 72.7%,”

Comment 2:

The introduction chapter should end with a paragraph indicating the purposefulness of the conducted research. Authors should clearly define the purpose of the work and formulate research hypotheses.

Response: The authors agree with this comment. The modified content is in Lines 101-106.

“Considering the aforementioned information, it is proposed that the direct contact of castor seedling roots may affect the early root cell morphology development, and the roots of castor seedlings may produce antioxidant enzymes resist heavy metal stress to ensure the normal growth. So, the purpose of this study was …”.

Comment 3:

Materials and method section is well described and correspond to the aim set out in the manuscript. The figures clearly presenting the obtained results with their appropriate interpretation.

Response: Thanks for this comment.

Comment 4:

How was the statistical calculations done?

Response: Thanks, the statistical calculations has been added in Lines 231-236.

 2.8 Statistical analysis

Data were analyzed by two-way ANOVA with Duncan’s test (p < 0.05) using SPSS 19 for Windows (SPSS Inc. IBM Corporation, Armonk, NY, USA). All the data are expressed as the mean ± SD of three replicates. The differences between different treatments were calculated using least significant different test. All the figures were per-formed using Origin Pro 9.”.

Comment 5:

The references are sufficient and necessary.

Response: Thanks.

Comment 6:

The paper needs some editorial corrections.

Response: Thanks. The whole manuscript has been revised based on your valable comments.

In line 20: “in space depth” was changed to “in different depth”.

In line 27: “in soil will” was changed to “in soils will”

In line 29: “adjust” was deleted; “changes of” was changed to “change”.

In line 38: “experts and” was deleted; “pay” was changed to “have being paid”

In line 39: “,” was deleted.

In line 69: “the” was added.

In line 70: “plant” was deleted.

In line 72: “involving” was added; “(include” was deleted.

In line 73: “and” was added; “,” was deleted.

In line 74: “, ect)” was deleted.

In line 78: “study, founding” was changed to “studies found”

In line 79: “sequencing” was changed to “sequence”

In line 95: “hydrogen peroxide ( )” was deleted.

In line 105: “ions” was deleted.

In line 186: “Zn380” was deleted.

In line 194: “.” was deleted.

In line 146: “concentration” was deleted.

In line 244: The font and size of fig.1 was revised.

In line 265: The font and size of fig.2 was revised.

In line 330: “increase” was changed to “increased”

In line 332: “that” was added.

In line 349: “stress proves castor” was changed to “stress, which proves that castor”.

In line 355: “is a strong evidence” was changed to “proves”.

In line 400: “was” was changed to “were”.

In line 406: “space depth” was changed to “different depths”.

Comment 7:

I recommend the publication of this manuscript in the Sustainability journal after minor revisions.

Response: Thanks. The whole manuscript has been revised based on your valable comments.

Reviewer 2 Report

The manuscript is interesting, but i'm not sure if the paper fits in the journal scope and has several issues that should be reviewed. The paper has several issues, including missing information and poor design.

L13. "Castor oil plants are used"

L14. "However, the interaction between heavy metals and castor seedling roots directly contacting heavy metals is unknown" I'm not sure... https://doi.org/10.1080/15226514.2021.1985959 https://doi.org/10.4314/jasem.v25i3.10 https://doi.org/10.1139/er-2020-0016 https://doi.org/10.1016/j.chemosphere.2020.126471 https://doi.org/10.1080/10889868.2014.979277 Try to avoid this kind of sentences.

L29. Cd and Zn are in the title and also in the abstract. Give the specie name (Ricinus communis) or another kind of keywords.

L33-35. Not only these elements...and Zn is needed but in very small ammounts.

L36-39. I'm not sure if use the Nemerrow index as a measure could be a good measure. Besides, how many soils or areas were assessed? This sections should be removed or improved. 

L41. "Ricinus communis L". Try to justify better why this specie. See https://doi.org/10.1139/er-2020-0016

L48. "Plant roots is most important to growth, including support and fixation of the above-ground parts, material storage, absorption" seems obvious... 

L56-88. This section is interesting, and I suggest focusing here and improve the previous sections. However, i suggest improving why the research was focused on roots and not also in the aerial part. The aerial part is also important for growth, including support, etc. 

L89-90. 

- "10 mesh sieve before use" Add mm size.

- Please give soil properties. This information is missing and seems vital because the authors use this soil as control... 

- Which kind of soil? Agricultural? Why not an artificial soil such as an OECD? 

L90-99. I think that this information is not needed. 

L102. Why 2 months? Were stabilized in the same conditions than the plant growth? 

L106-107. Why not an artificial soil as a control?

L107-111. These sentences should be improved.

L111. How many replicates?

L112. H2O2? Why? Usually, bleach is applied, but not H2O2!

L114. "The castor seeds were planted on 1st September." Not needed

L101-118. Growth conditions are missing (temperature, light and dark hours, etc)

L145. Dry weight is done at 60C, not 75. 

L199. Statistical treatment information is missing.

L200. All results and discussion sections need several improvements. We can not understand the behaviour of the metals without the background values of the studied soils. According to your values in Fig 1, Control seeds have more Cd than Cd0... seems that you control soil is not a good control soil. 

Author Response

The manuscript is interesting, but i'm not sure if the paper fits in the journal scope and has several issues that should be reviewed. The paper has several issues, including missing information and poor design.

Response: The comments from Reviewer #2 are appreciated. Please note: line numbers in the Reviewer’s comments refer to the last version of the manuscript, while the line numbers in the Authors’ answers refer to the current revised manuscript.

Comment 1:

L14. "However, the interaction between heavy metals and castor seedling roots directly contacting heavy metals is unknown" I'm not sure...

https://doi.org/10.1080/15226514.2021.1985959 https://doi.org/10.4314/jasem.v25i3.10

https://doi.org/10.1139/er-2020-0016https://doi.org/10.1016/j.chemosphere.2020.126471 https://doi.org/10.1080/10889868.2014.979277 Try to avoid this kind of sentences.

Response: The authors agree with this comment. The revised sentence is in line 14.

“However, only limited studies addressed the interaction between heavy metals and castor seedling roots directly contacting heavy metals”.

Comment 2:

L29. Cd and Zn are in the title and also in the abstract. Give the specie name (Ricinus communis) or another kind of keywords.

Response: The authors agree with this comment. The new keyword was added in line 31.

“heavy metals”.

Comment 3:

L33-35. Not only these elements...and Zn is needed but in very small ammounts.

Response: The authors agree with this comment. The part has been restated in line 36.

“The excessive Cadmium (Cd) and Zinc (Zn) induce serious risks…”

Comment 4:

L36-39. I'm not sure if use the Nemerrow index as a measure could be a good measure. Besides, how many soils or areas were assessed? This sections should be removed or improved. 

Response: Thanks, the part has been added in Lines 39-45.

“The Ministry of Land and Resources Report issued the national soil contamination survey results and reported that 16% of the surveyed Chinese soil samples, 6.3 million km2, were contaminated [1]. A previous study reported that average concentration of Zn in farmland soils of China is 4.7 times higher than the soil secondary environmental quality standard through investigating 5000 soil samples from 1781 farmlands [2].”

Comment 5:

L41. "Ricinus communis L". Try to justify better why this specie. See https://doi.org/10.1139/er-2020-0016

Response: Thanks for your advice. More supplementary notes have been added in lines 52-54.

“Castor is an important oil seed crop worldwidely and no vegetative part of castor is consumed by human beings or cattle. Therefore, it is safe for the purpose of soil phytoremediation [6]”.

Comment 6:

L48. "Plant roots is most important to growth, including support and fixation of the above-ground parts, material storage, absorption" seems obvious... 

Response: The authors agree with this comment. The revised sentences is in lines 60-62.

“Plant roots can support and fixate the above-ground parts. It plays an important role in material storage, absorption of nutrients such as water and inorganic salts, and synthesis of substances such as amino acids and hormones”.

Comment 7:

L56-88. This section is interesting, and I suggest focusing here and improve the previous sections. However, i suggest improving why the research was focused on roots and not also in the aerial part. The aerial part is also important for growth, including support, etc.

Response: Thanks for your advice. More supplementary notes have been added in lines 78-84.

“One of our previous study found that the sequence for Zn and Cd accumulation in castor seedling tissues was root > leaf > stem and root > stem > leaf, respectively, indicating that castor seedling root played an important role in accumulating Zn and Cd [3]. … However, there are very few studies addressing the interaction between heavy metals and castor seedling roots directly contacting heavy metals”.

Comment 8:

L89-90. 

- "10 mesh sieve before use" Add mm size.

- Please give soil properties. This information is missing and seems vital because the authors use this soil as control... 

- Which kind of soil? Agricultural? Why not an artificial soil such as an OECD? 

Response: Thanks. In order to get close to the natural soil, non-polluted agricultural soil was collected from a farmland at the campus of Shanghai University, and then Zn and Cd were dosed into the soil before use in this study. This part was added in lines 111-130.

“and the experimental soils were taken from a non-polluted farmland at the campus of Shanghai University (Shanghai, China). The collected soil was air-dried naturally and passed through 10 mesh sieve (0.25mm) before use. Approximately 5 g of soil was collected to determine the background values of the soil, including pH, TN, TP, TOM, TK, Zn and Cd. Additionally, after soil aging, the concentration of TK, Zn and Cd in soils was measured. The physical and chemical properties of the experimental soil are shown in Table 1”.

Table 1

Physical and chemical parameters of experimental soil.

Physical or chemical parameters

Soil background

Added heavy metals

(mg/kg soil)

After adding heavy metals

pH

8.03

0

--

TOC

13.63 g/kg

0

--

TN

1.69 g/kg

0

--

TP

0.93 g/kg

0

--

TK

19.2 g/kg

0

19.1 g/kg

Cd

0.18 mg/kg

25

23.54 mg/kg

5

5.4 mg/kg

0.5

0.46 mg/kg

0

0.2 mg/kg

Zn

73.25 Mg/kg

380

448.1 mg/kg

0

93 mg/kg

Comment 9:

L90-99. I think that this information is not needed. 

Response: The authors agree with this comment. This part was deleted.

Comment 10:

L102. Why 2 months? Were stabilized in the same conditions than the plant growth?

Response: Thanks, The supplement is in lines 133-134. There are two reasons for 2 months:1) heavy metal aging usually for 1 months, the longer the aging time, the more stable the form of heavy metals[4];2) Combined with the previous experience of the laboratory.

Comment 11:

L106-107. Why not an artificial soil as a control?

Response: Thanks. Compared with the artificial soil, the in-site non-polluted soil with heavy metal addition is more close to the real situation. Additionally, the difference between control and treatment groups were the presence of heavy metals. By comparing the control and the treatment groups, the objectives of this study can be achived.

Reviewer 3 Report

The manuscript entitled Response of castor seedling roots to combined pollution of Cd and Zn in soils” discusses the interaction between heavy metals and castor seedling roots. In this study, physiological response of castor seedling roots to Cd and Zn stress and the change of trace elements in rhizosphere and non-rhizosphere soils were investigated. Results showed that with the increase of Cd concentration, the accumulation of Zn in roots decreased, indicating a competition between Cd and Zn accumulation. The purpose of this study was to investigate the effects of castor seedling roots on Cd 82 and Zn ions and other trace elements, and the response in morphology and physiology of castor seedling roots to combined pollution of Cd and Zn.

The below revisions are recommended:

1.       Lines#144 & 204: How many (either in number or in weight) seedlings were taken in each group?

2.       I recommend ICP-MS determinations of Cd and Zn separately for the roots and the aerial part of the seedlings. This comparative study will be helpful to understand the accumulation of Cd and Zn by the roots and the aerial part of the seedlings.

3.       The manuscript requires thorough revision of spellings and grammar. Also, uniformity (font and size) should be maintained throughout the manuscript including the schemes and figures.

Author Response

The manuscript entitled “Response of castor seedling roots to combined pollution of Cd and Zn in soils” discusses the interaction between heavy metals and castor seedling roots. In this study, physiological response of castor seedling roots to Cd and Zn stress and the change of trace elements in rhizosphere and non-rhizosphere soils were investigated. Results showed that with the increase of Cd concentration, the accumulation of Zn in roots decreased, indicating a competition between Cd and Zn accumulation. The purpose of this study was to investigate the effects of castor seedling roots on Cd 82 and Zn ions and other trace elements, and the response in morphology and physiology of castor seedling roots to combined pollution of Cd and Zn.

Response: The comments from Reviewer #3 are appreciated. Please note: line numbers in the Reviewer’s comments refer to the last version of the manuscript, while the line numbers in the Authors’ answers refer to the current revised manuscript.

Comment 1:

 Lines#144 & 204: How many (either in number or in weight) seedlings were taken in each group?

Response: Thanks.

For line 144, weight was added in line 177.

“They were dried in an weight oven at 75 ℃, then grinded into fine powder which taken 0.1g powder and digested using HNO3/HClO4 (5:1, v/v)”.

For line 204, please watch table 1(new added). They are added in lines 128-130.

“Table 1

Physical and chemical parameters of experimental soil.”

Physical or chemical parameters

Soil background

Added heavy metals

(mg/kg soil)

After adding heavy metals

pH

8.03

0

--

TOC

13.63 g/kg

0

--

TN

1.69 g/kg

0

--

TP

0.93 g/kg

0

--

TK

19.2 g/kg

0

19.1 g/kg

Cd

0.18 mg/kg

25

23.54 mg/kg

5

5.4 mg/kg

0.5

0.46 mg/kg

0

0.2 mg/kg

Zn

73.25 Mg/kg

380

448.1 mg/kg

0

93 mg/kg

Comment 2:

 I recommend ICP-MS determinations of Cd and Zn separately for the roots and the aerial part of the seedlings. This comparative study will be helpful to understand the accumulation of Cd and Zn by the roots and the aerial part of the seedlings.

Response: Thanks for your advice. One of our previous study compared aboveground and underground accumulation of Cd and Zn, which is supplemented in lines 371-374.

One of our previous studies found that the sequence of Zn and Cd concentrations existing in castor seedling tissues was root > leaf > stem and root > stem > leaf, respectively, indicating that castor seedling root played an important role in accumulating Zn and Cd[3].

Comment 3:

The manuscript requires thorough revision of spellings and grammar. Also, uniformity (font and size) should be maintained throughout the manuscript including the schemes and figures.

Response: Thanks, The whole manuscript has been revised based on your valable comments.

In line 20: “in space depth” was changed to “in different depth”.

In line 27: “in soil will” was changed to “in soils will”

In line 29: “adjust” was deleted; “changes of” was changed to “change”.

In line 38: “experts and” was deleted; “pay” was changed to “have being paid”

In line 39: “,” was deleted.

In line 69: “the” was added.

In line 70: “plant” was deleted.

In line 72: “involving” was added; “(include” was deleted.

In line 73: “and” was added; “,” was deleted.

In line 74: “, ect)” was deleted.

In line 78: “study, founding” was changed to “studies found”

In line 79: “sequencing” was changed to “sequence”

In line 95: “hydrogen peroxide ( )” was deleted.

In line 105: “ions” was deleted.

In line 186: “Zn380” was deleted.

In line 194: “.” was deleted.

In line 146: “concentration” was deleted.

In line 244: The font and size of fig.1 was revised.

In line 265: The font and size of fig.2 was revised.

In line 330: “increase” was changed to “increased”

In line 332: “that” was added.

In line 349: “stress proves castor” was changed to “stress, which proves that castor”.

In line 355: “is a strong evidence” was changed to “proves”.

In line 400: “was” was changed to “were”.

In line 406: “space depth” was changed to “different depths”.

  1. Yang, Q., et al., A review of soil heavy metal pollution from industrial and agricultural regions in China: Pollution and risk assessment. Sci Total Environ, 2018. 642: p. 690-700.
  2. Li, Z., et al., A review of soil heavy metal pollution from mines in China: pollution and health risk assessment. Sci Total Environ, 2014. 468-469: p. 843-53.
  3. He, C., et al., Phytoremediation of soil heavy metals (Cd and Zn) by castor seedlings: Tolerance, accumulation and subcellular distribution. Chemosphere, 2020. 252: p. 126471.
  4. Wu, C. and C. He, Interaction effects of oxytetracycline and copper at different ratios on marine microalgae Isochrysis galbana. Chemosphere, 2019. 225: p. 775-784.

Round 2

Reviewer 2 Report

The manuscript has been improved since the last submission, and the manuscript should be more interesting with data from the aerial section. - Besides, in this kind of paper, the plant extractable contents (e.g. DTPA, CaCl2, EDTA-extraction) should be measured since total content is not a good measure to assess the impact of metals in plants. Cd and Zn have a low solubility under a soil pH of 8.03!! This information is missing and should be provided. 

However, I have some questions that should be improved.

- Table 1. Remove "0" from added HM column for physicochemical parameters. If possible, please improve the information for 0, 0.5, 5, 25 mg kg and 0/380 mg kg. Add a different column or line with the different treatments, or even in the text, but in the present version seems a bit confusing. Probably, this table could be removed since this information was indicated in section 2.2.

- Did you measure the soil parameters after adding HM? 

- Besides, in this kind of paper, the plant extractable contents (e.g. DTPA, CaCl2, EDTA-extraction) should be measured since total content is not a good measure to assess the impact of metals in plants. Cd and Zn have a low solubility under a soil pH of 8.03!! This information is missing and should be provided. 

- L229-230 and 316-317 "and the same letter means no significant difference (p>0.05)" This is not needed. Only information for different letters should be provided.

-L244-245. Statistical information is missing in Figure 2.

- L247. Sato or Akai? Reference #30 is Sato. Please, revise this kind of questions throughout the manuscript.

- L276-291. This figure legend seems too bigger. Reduce this figure legend, and probably part of the text of this figure legend should be written in the manuscript.

- L360-367. The discussion section for oxidative stress is scarce

- References are not according to the journal guidelines. https://www.mdpi.com/journal/sustainability/instructions

Author Response

Reviewer

The manuscript has been improved since the last submission, and the manuscript should be more interesting with data from the aerial section. - Besides, in this kind of paper, the plant extractable contents (e.g. DTPA, CaCl2, EDTA-extraction) should be measured since total content is not a good measure to assess the impact of metals in plants. Cd and Zn have a low solubility under a soil pH of 8.03!! This information is missing and should be provided. However, I have some questions that should be improved.

Response: The comments from Reviewer are appreciated. Please note: line numbers in the Reviewer’s comments refer to the last version of the manuscript, while the line numbers in the Authors’ answers refer to the current revised manuscript.

Comment 1:

Table 1. Remove "0" from added HM column for physicochemical parameters. If possible, please improve the information for 0, 0.5, 5, 25 mg kg and 0/380 mg kg. Add a different column or line with the different treatments, or even in the text, but in the present version seems a bit confusing. Probably, this table could be removed since this information was indicated in section 2.2.

Response: The authors agree with this comment. The table 1 was improved, and the modified table 1 is in Lines 104-109.

Comment 2:

Did you measure the soil parameters after adding HM? 

Response: The contents of Cd and Zn in soils were measured after adding heavy metals, and the results are showun in Table 1 (Lines 107-109).

Comment 3:

Besides, in this kind of paper, the plant extractable contents (e.g. DTPA, CaCl2, EDTA-extraction) should be measured since total content is not a good measure to assess the impact of metals in plants. Cd and Zn have a low solubility under a soil pH of 8.03!! This information is missing and should be provided. 

Response: Thanks for this comment. Diethylenetriaminepentaacetic acid (DTPA) and ethylenediamine tetraacetic acid (EDTA) have been widely used due to their ability to form very stable, water-soluble and well-defined complexes with metal cations. Calcium chloride, soil background electrolyte solutions, are also frequently adopted as extractants for bioavailability prediction. But this study did not focus on bioavailability prediction, and the purpose of this study was to investigate effects of castor seedling roots on Cd and Zn ions and other trace elements, the response in morphology and physiology of castor seedling roots to combined pollution of Cd and Zn. The content of Cd and Zn in soil was measured not in water. So the content of Cd and Zn of pore water of soil don’t be measured. In the future study, this comment will be considered. Thanks.

Comment 4:

L229-230 and 316-317 "and the same letter means no significant difference (p>0.05)" This is not needed. Only information for different letters should be provided.

Response: Thanks, the "and the same letter means no significant difference (p>0.05)" was removed in lines 231-232, 269 and 319.

Comment 5:

L244-245. Statistical information is missing in Figure 2.

Response: The authors agree with this comment. The “did not change significantly” was changed to “did little change” in line 238.

Comment 6:

L247. Sato or Akai? Reference #30 is Sato. Please, revise this kind of questions throughout the manuscript.

Response: Thanks.

In line114: the references “(Wan et al.,2018, Jin et al., 2019)” in line 136 was removed and the references [27,28] was added in line 116.

In line 247: the “, Akai” and references in line 271 was removed and was added in line 249.

In line 331: the “, Ma” and references in line 331 was removed and was added in line 332.

Comment 7:

L276-291. This figure legend seems too bigger. Reduce this figure legend, and probably part of the text of this figure legend should be written in the manuscript.

Response: Thanks for this comment. This figure legend has been revised in line 244.

Comment 8: L360-367. The discussion section for oxidative stress is scarce

Response: Thanks, more discussion have been added in lines 366-369,375-378, respectively.

“The SOD enzyme activity in treatment groups was significantly higher than that in the control group, suggesting that the addition of Cd and Zn stimulates the antioxidant enzyme system of castor seedling to cope with the root damage caused by external environment.”

“In this study, with the increase of the Cd content the MDA content also increased, suggesting that the addition of Cd and Zn may damage the cells in the roots of castor seedlings, and the cell morphology morphological changes of roots may be related to this.”

Comment 9: References are not according to the journal guidelines. https://www.mdpi.com/journal/sustainability/instructions

Response: Thanks. The format of references has been fully corrected which were in lines 412-614.

Round 3

Reviewer 2 Report

All comments have been addressed. However, i think that available contents should be measured, and the paper should have other focus than the present version.

Author Response

Response to Reviewer Comments

The authors appreciate your comments. We have revised this manuscript based on your valuable comments in the first two rounds. Hereby we would like to clarify the knowledge gap, main research question and this study’s contents frist and then re-answer your two main comments from the second round.

Castors are used to remediate heavy metal polluted soils due to their good accumulation capacity of heavy metals. However, only limited studies addressed the interaction between heavy metals and castor seedling roots directly contacting heavy metals. In this study, physiological response of castor seedling roots to Cd and Zn stress and the change of trace elements in rhizosphere and non-rhizosphere soils were investigated. The morphological changes of external and internal cells of seedling roots under different degrees of soil pollution were observed, as well as the distribution of micronutrient elements in the external growth environment and the changes of internal physiological indexes.

Comment 2:

Did you measure the soil parameters after adding HM? 

Response: The contents of metal (Cd and Zn) and nutrient elements (K, P, Fe, Mn) in soils after adding heavy metals were measured, and the results are showun in Table 1 (Lines 107-109) and figure 2 (in line 242), respectively.

Comment 3:

Besides, in this kind of paper, the plant extractable contents (e.g. DTPA, CaCl2, EDTA-extraction) should be measured since total content is not a good measure to assess the impact of metals in plants. Cd and Zn have a low solubility under a soil pH of 8.03!! This information is missing and should be provided. 

Response: Thanks for this comment. Diethylenetriaminepentaacetic acid (DTPA) and ethylenediamine tetraacetic acid (EDTA) have been widely used due to their ability to form very stable, water-soluble and well-defined complexes with metal cations. Calcium chloride, soil background electrolyte solutions, are also frequently adopted as extractants for bioavailability prediction. This study does not focus on bioavailability prediction, and the purpose of this study was to investigate effects of castor seedling roots on Cd and Zn ions and other trace elements, the response in morphology and physiology of castor seedling roots to combined pollution of Cd and Zn. 

In this paper, the change of heavy metal speciations (including the content of Cd and Zn which them have a low solubility under a soil pH of 8.03) in soil was not measured, as you said, the total metal content is not a good evaluation index, the change of heavy metal form can reflect the biological toxicity of soil. We thanks for your advice. In the follow-up study, this comment will be considered.
